# A Study on the Influence of Suspended Matter by the Foundation Construction of Different Offshore Wind Turbines in the Taiwan Sea Area

Sung-Shan Hsiao [1], Hsing-Yu Wang [2] and Yun-Chih Chiang [3],*

1. Department of Harbor and River Engineering, College of Engineering, National Taiwan Ocean University, Keelung 20224, Taiwan; sshsiao@ntou.edu.tw
2. Department of Shipping Technology, College of Maritime, National Kaohsiung University of Science and Technology, Kaohsiung 805, Taiwan; hywang05@nkust.edu.tw
3. Center for General Education, College of Education & Communication, Tzu Chi University, Hualien 97004, Taiwan
* Correspondence: ycchiang@mail.tcu.edu.tw; Tel.: +886-3-8565301 (ext. 5517)

**Abstract:** The objective of this study was to adequately examine potential wave fields, flow fields, and suspended load changes in different wind turbine foundations. Accordingly, this study applied the hydrodynamic model to simulate waves, currents, and suspended load in the study area. The simulation results are based on the assumption that dredging and rubble bed trimming were performed for 8 h and that the per foundation setting operation was completed in 2 h. The influence on the tripile and jacket was larger than that on the monopile, and the influence time was longer. However, due to the influence of tidal currents on the sea, the suspended load also became more acceptable than the initial concentration. From a macroscopic perspective, the different foundations did not sufficiently affect the study area. From a microscopic perspective, changes in the suspended load were only limited to areas surrounding the piles after the installation of the wind turbines.

**Keywords:** wind turbine foundation; hydrodynamic model; suspended load





## 1. Introduction

Due to the increasing environmental awareness in recent years, people have begun to consider the problems of air pollution and nuclear waste treatment caused by fossil fuels and nuclear power generation, respectively. Therefore, determining economically beneficial and environmentally friendly power generation methods has become an imperative task for Taiwan. Over the past 10 years, Taiwan has been actively developing terrestrial wind farms. Currently, a total of 24 wind farms have been established on the island. Nevertheless, favorable locations for developing wind farms have been exhausted, and wind turbine generators because considerable noise engendered by wind shear; therefore, establishing offshore wind farms will become a future trend for Taiwan.

Taiwan is a nation surrounded by the ocean and has ideal conditions for the development of marine applications. In recent years, the needs for economic development, the improvement in marine clean energy technology, the rise in marine recreational activities, the development of new types of coasts and oceans, and the rise in marine environmental awareness highlight the problems of space use. Based on the issues of global warming and clean energy demand in recent years and under the pressure of abolishing nuclear energy and reducing carbon emissions, offshore wind power has become one of the options for Taiwan's energy transformation. The offshore wind field in the Taiwan Strait has excellent resources. For example, 4C Offshore: Marine Consultants uses the observation data of the average wind speed to show that 13 of the best observation sites in the world are located on the western coast of Taiwan [1]. According to the selection results of offshore wind power planning site selection [2], the Changhua sea area in western Taiwan has a capacity

of as high as 62.6% for offshore wind power installations, ranking first in Taiwan's offshore wind power generation capacity distribution area.

The layout of offshore structures is closely related to sedimentation because the layout of engineering structures changes the sediment bed dynamic conditions of the original flow field and wave field. In order to avoid the influence of the wind turbine's wake, the layout of the generator is generally set to use the distance of up to 4 times the parallel wind direction and the distance of 10 times the vertical wind direction as the interval. This arrangement is mainly used to avoid the kinetic energy loss caused by downstream wind turbines [3–5]. In addition, referring to the relevant environmental protection specifications of the Environmental Impact Assessment Inquiry System for the development of offshore wind farms in the sea area around Taiwan, with the future development of large-scale wind turbines, the offshore wind turbines planned by Taiwan's offshore wind power developers are separated by about 1 km. The development utilized in the comprehensive planning of ocean space was carried out in the long term, prioritizing public interest regarding the use of the sea area for offshore wind power [6,7].

In order to take into account, the comprehensive evaluation of the use of space in the sea area, this study uses the hydrodynamic model under the influence of wave–current interaction to simulate the wave–current pattern around the structure. In addition, studies on the impact of offshore wind farms on marine ecology have mostly focused on the impact of fish, birds, and cetaceans, and there are relatively few studies on the impact on the marine environment and marine basic productivity. Among them, the three most important environmental factors affecting the primary productivity of the ocean are the abundance of light in the marine environment, the stability of water bodies, and the abundance of seawater nutrients. Due to the construction of offshore wind turbines which may increase suspended loads in the seawater and obscure the light in the sea, this effect may weaken the photosynthesis ability of phytoplankton in the seawater, leading to a decline in primary productivity. Accordingly, this study simulated the suspended load changes occurring after the installation of different wind turbine foundations. Numerical modeling was performed to examine the influence of the offshore wind farm set on its neighboring ocean environment. The results can serve as a reference for subsequent engineering construction activities and long-term geomorphological change analysis.

## 2. Methodology

### 2.1. Hydrodynamic Model

In this study, the tidal effect is caused by the tide, which is added to the hydrodynamic model. The governing equation of the two-dimensional hydrodynamic model can be derived through the following process [8–12]: first, the Navier–Stokes equation is simplified by assuming an incompressible fluid and hydrostatic pressure distribution. The conservation of mass and momentum equations are then acquired on the basis of a depth integral equation and appropriate boundary conditions:

$$\frac{\partial \eta}{\partial t} + \frac{\partial}{\partial x}[U(h + \eta)] + \frac{\partial}{\partial y}[V(h + \eta)] = 0 \tag{1}$$

$$\frac{\partial U}{\partial t} + U\frac{\partial U}{\partial x} + V\frac{\partial U}{\partial y} = fv - g\frac{\partial \eta}{\partial x} + \frac{1}{\rho}\left(\frac{\partial \tau_{xx}}{\partial x} + \frac{\partial \tau_{yx}}{\partial y}\right) + \frac{1}{\rho(h+\eta)}(\tau_{sx} - \tau_{bx}) - \frac{1}{\rho(h+\eta)}\left(\frac{\partial S_{xx}}{\partial x} + \frac{\partial S_{yx}}{\partial y}\right) \tag{2}$$

$$\frac{\partial V}{\partial t} + U\frac{\partial V}{\partial x} + V\frac{\partial V}{\partial y} = fv - g\frac{\partial \eta}{\partial x} + \frac{1}{\rho}\left(\frac{\partial \tau_{xy}}{\partial x} + \frac{\partial \tau_{yy}}{\partial y}\right) + \frac{1}{\rho(h+\eta)}(\tau_{sy} - \tau_{by}) - \frac{1}{\rho(h+\eta)}\left(\frac{\partial S_{xy}}{\partial x} + \frac{\partial S_{yy}}{\partial y}\right) \tag{3}$$

where $\eta$ is the water surface elevation (m) [13]; $h$ represents the distance from the static water level to the bed (m); $g$ is the gravitational constant (m/s$^2$); $U$ and $V$ represent the mean current velocity (m/s) of the water depth in the fixed coordinates of the $x$ and $y$ axes:

$$U = \left(\frac{1}{h+\eta}\right)\int_{-h}^{\eta} u\,dz, \quad V = \left(\frac{1}{h+\eta}\right)\int_{-h}^{\eta} v\,dz \tag{4}$$

Shear stresses $\tau_{xx}$, $\tau_{xy}$, $\tau_{yx}$, and $\tau_{yy}$ include viscous stress caused by fluid viscosity and Reynold's stress caused by turbulent effects. As the value of viscous stress compared with Reynold's stress is very small, viscous stress is ignored generally, and only Reynolds stress is considered to represent the momentum exchange between fluids:

$$\tau_{xx} = \rho E_v \frac{\partial U}{\partial x}, \quad \tau_{xy} = \rho E_v \frac{\partial U}{\partial y}, \quad \tau_{yx} = \rho E_v \frac{\partial V}{\partial x}, \quad \tau_{yy} = \rho E_v \frac{\partial V}{\partial y} \tag{5}$$

The vortex viscosity coefficient $E_v$ is obtained from the semi-empirical formula of the Prandtl mixing length theory [14]:

$$E_v = \frac{k_v \sqrt{g}(d+h)\sqrt{U^2+V^2}}{6C_c} \tag{6}$$

The sea surface wind shear components $\tau_{sx}$ and $\tau_{sy}$ are the components of the sea surface wind shear in the $x$ and $y$ directions [15]:

$$\tau_{sx} = \rho k_w W^2 \cos a; \quad \tau_{sy} = \rho k_w W^2 \sin a \tag{7}$$

$$k_w = \begin{cases} 1.2 \times 10^{-5} & , \quad W \leq W_c \\ 1.2 \times 10^{-6} + 2.25 \times 10^{-6}\left[1 - \frac{W_c}{W}\right]^2 & , \quad W > W_c \end{cases} \tag{8}$$

The bottom friction stresses $\tau_{bx}$ and $\tau_{by}$ are the components in the $x$ and $y$ directions [16]:

$$\tau_{bx} = \rho E_r U \sqrt{U^2+V^2}; \quad \tau_{by} = \rho E_r V \sqrt{U^2+V^2} \tag{9}$$

where the coefficient of the bottom friction is $F_r = g/C_c^2$.

$S_{xx}$, $S_{xy}$, $S_{yx}$, and $S_{yy}$ represent the components of the radiation stress caused by waves (kg/ms$^2$), which can be estimated using the method proposed by [8]:

$$\begin{bmatrix} S_{xx} & S_{xy} \\ S_{yx} & S_{yy} \end{bmatrix} = \frac{\rho g H_a^2}{g} \begin{bmatrix} n\left(1+\cos^2\theta\right) - \frac{1}{2} & \frac{n}{2}\sin 2\theta \\ \frac{n}{2}\sin 2\theta & n\left(1+\sin^2\theta\right) - \frac{1}{2} \end{bmatrix} \tag{10}$$

where $\overline{E}$ is the total wave energy per unit time and area of section. Under airy wave theory,

$$\overline{E} = \frac{\rho g H^2}{8} \tag{11}$$

The boundary conditions of the hydrodynamic model are shown in Figure 1. The water level change includes the level rise and fall caused by waves and tides.

The tide is a sine function changing from right to left, $T_L$ is the time difference when the tide reaches the left and right boundaries, and $T_{start}$ is the phase difference:

Left boundary:

$$\eta_L = A_L^t \sin\left[\frac{2\pi}{T_t}(t + T_t + T_{start})\right], \quad T_t = \frac{L_y}{\sqrt{gh_{Max}}} \tag{12}$$

Right boundary:

$$\eta_R = A_R^t \sin\left[\frac{2\pi}{T_t}(t + T_{start})\right] \tag{13}$$

Offshore boundary:

$$\eta_0 = \left[ A_R^t + \left( A_L^t + A_R^t \right) \left( \frac{N_y - j}{N_y - 1} \right) \right] \sin \left\{ \frac{2\pi}{T_t} \left[ t + T_t \left( \frac{N_y - j}{N_y - 1} \right) + T_{start} \right] \right\} \tag{14}$$

The water level $\overline{\xi}$ caused by waves is based on [13], ignoring the reflection effect. Equation (15) is the water level descent outside the surf zone, and Equation (16) is the water level uplift in the surf zone:

$$\overline{\xi_d} = -\frac{H^2}{8} \frac{k}{\sinh(2kh)} (\cos \theta)^{2/3} \tag{15}$$

$$\frac{d\overline{\xi_u}}{dx} = -K \frac{dh}{dx}, \quad K = \frac{1}{1 + (8/3\gamma^2)} \tag{16}$$

The left boundary of the velocity is shown in Equation (17), the right boundary is shown in Equation (18), the offshore boundary is shown in Equation (19), and the longshore boundary is shown in Equation (20).

$$U_{j=1} = U_{j=2}, \quad \left( \frac{\partial V}{\partial y} \right)_{j=1} = 0 \tag{17}$$

$$U_{j=NY} = U_{j=NY-1}, \quad \left( \frac{\partial V}{\partial y} \right)_{j=NY} = 0 \tag{18}$$

$$U_{j=NX} = U_{j=NX-1}, \quad \left( \frac{\partial U}{\partial y} \right)_{j=NX} = 0 \tag{19}$$

$$U = 0, \quad V = 0 \tag{20}$$

The stability of the hydrodynamics calculation must satisfy $\Delta t \leq 2\Delta s / \sqrt{gh_{Max}}$, where $\Delta s$ is the grid size. The maximum value of two adjacent time steps is less than the allowable error with Equation (21), and then the calculation of the next time step can be performed.

$$\begin{aligned} Max \left( \eta_{ij}^{k+1} - \eta_{ij}^k \right) \leq \varepsilon_\eta \eta_{ij}^k \quad &, \quad \varepsilon_\eta = 0.0001 \\ Max \left( U_{ij}^{k+1} - U_{ij}^k \right) \leq \varepsilon_U U_{ij}^k \quad &, \quad \varepsilon_U = 0.0001 \\ Max \left( V_{ij}^{k+1} - V_{ij}^k \right) \leq \varepsilon_V V_{ij}^k \quad &, \quad \varepsilon_V = 0.0001 \end{aligned} \tag{21}$$

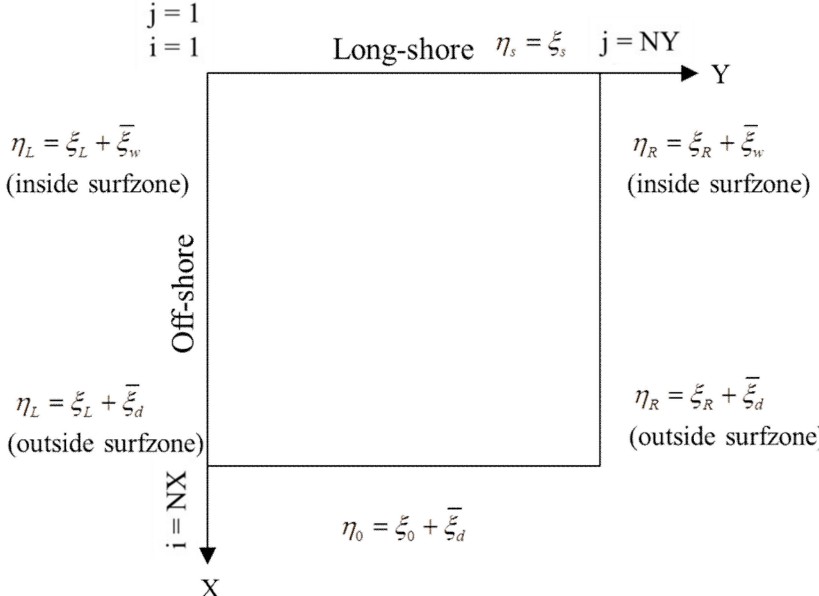

**Figure 1.** Hydrodynamic model boundary condition diagram.

## 2.2. Wave Model

The data required to calculate radiation stress are the wave height, period, and direction, which can be obtained from various types of wave models addressing dissimilar physical problems. When both waves and tides are present, there is substantial diversity in both the spatial and temporal scales; thus, if waves and tides are considered to be on the same space–time scale, obtaining practical engineering applications for large sea areas is difficult. Accordingly, the calculation efficiency can be substantially improved if the force exerted by the wave field is regarded as steady over a certain period of time, thus ignoring instantaneous changes in wave motion. Considering refraction, diffraction, and wave breaking caused by the transmission of deep-water waves to shallow-water areas, this study's calculation of wave patterns was carried out by using the mild slope equation of the current effect [17]:

$$\frac{D^2\varphi}{Dt^2} + \left(\nabla \cdot \vec{U}\right)\frac{D\varphi}{Dt} - \nabla \cdot (CC_g \nabla \varphi) + \left(\sigma^2 - k^2 CC_g\right)\varphi = 0 \tag{22}$$

where $\vec{U}$ is the ambient current, $\nabla$ is the horizontal gradient operator, $\varphi$ is the two-dimensional velocity potential, $k$ is the wave number, $C$ and $C_g$ are the phase and group speed of the waves, and $\sigma$ is the dispersion relation given by $\sigma^2 = gk\tan kh$. Under the assumption of an irrotational field, with single-frequency linear surface waves, the potential energy of a wave can be expressed as follows:

$$\varphi\left(\vec{x}, \vec{y}, z, t\right) = f(z, h)\varphi\left(\vec{x}, \vec{y}, t\right) \tag{23}$$

where $f(z, h) = \frac{\cosh[k(h+z)]}{\cosh kh}$.

In a single periodic harmonic motion, Equation (22) can be rewritten as follows:

$$\varphi\left(\vec{x}, \vec{y}, t\right) = \text{Re}\left\{ae^{is}e^{i\omega t}\right\} \tag{24}$$

The following expression can be obtained by substituting Equation (24) into Equation (22) for the real part and imaginary part:

$$\frac{1}{aCC_g}\left\{\left(\vec{U} \cdot \nabla a\right)\left[\left(\vec{U} \cdot \nabla\right) + \left(\nabla \cdot \vec{U}\right)\right]\right\} - \frac{1}{a}\left[\nabla^2 a + \frac{1}{CC_g}\left(\nabla CC_g \cdot \nabla a\right)\right] - k^2 + |\nabla s|^2 = 0 \tag{25}$$

$$\nabla \cdot \left[a^2\sigma(U + C_g)\right] = 0 \tag{26}$$

Equations (25) and (26) are the equations of motion for wave interactions before breaking waves. When the current velocity $\vec{U}$ is known, it solves the system of linear equations in two unknown parabolic simultaneous equations and obtains the amplitude $a(x, y)$ and the wave numbers $|\nabla s|$. When $\vec{U} = 0$, Equations (25) and (26) become:

$$\frac{1}{a}\left\{\frac{\partial^2 a}{\partial x^2} + \frac{\partial^2 a}{\partial y^2} + \frac{1}{cc_g}\left[\nabla a \cdot \nabla(cc_g)\right]\right\} + k^2 - |\nabla s|^2 = 0 \tag{27}$$

$$\nabla \cdot \left[a^2 CC_g \nabla s\right] = 0 \tag{28}$$

where $a$ represents the wave amplitude (*m*) and $S$ represents the phase function. RCPWAVE solves the mild slope equation of the parabolic type in a stable and fast manner with minimum computational calculation; hence, the model is reasonably applicable for making engineering application calculations for large sea areas.

In addition, energy is dissipated in the surf zone, and the energy expression of Equation (26) must be modified. Based on energy flux theory, ignore the effect of bottom friction [18]:

$$\frac{d(EC_g)}{dx} = -\varepsilon, \quad \varepsilon = \frac{1}{2}\rho V_e(kH_B)^2, \quad V_e = V_{eB}\left(\frac{\frac{H_B}{2} - c'h_B}{\gamma'h_B}\right),$$

$$V_{eB} = \frac{5S_Bg}{8k_B\rho} \frac{1}{\sqrt{1-C_0}}, \quad S_B = \frac{\tan\beta}{1+\frac{3r'^2}{2}} \tag{29}$$

where $c'$ is the ratio of the radiation to the water depth of the recovery zone. According to [18], $c' = 0.17$ when the wave recovery zone is not obvious in a gentle slope.

In the area of wave–current interaction, the energy dissipated by the nearshore current within the surf zone is small and negligible; therefore, the energy amplitude expression according to Equation (29) can be expressed as follows:

$$\nabla \cdot \left[\frac{E}{\sigma}\left(\vec{U} + C_g\right)\right] = -\frac{5}{16}\frac{\rho g^2 k_B}{\sigma^2}\frac{\tan\beta}{1+\frac{3r'^2}{2}}\frac{1}{\sqrt{1-C_0}}\sqrt{\frac{\frac{H_B}{2}-c'h_B}{r'h_B}}(H_B)^2 \tag{30}$$

With Equation (29), the energy in the surf zone is expressed, and Equation (30) is modified as follows in Equation (31).

$$\nabla \cdot \left[a^2\sigma\left(\vec{U} + C_g\right)\right] = \nabla \cdot \left[\frac{2g}{\rho}\frac{E}{\sigma}\left(\vec{U} + C_g\right)\right]$$

$$= -\frac{5}{8}\frac{g^2 k_B}{\sigma}\frac{\tan\beta}{1+\frac{3r'^2}{2}}\frac{1}{\sqrt{1-\frac{c'}{r'}}}\sqrt{\frac{\frac{H_B}{2}-c'h_B}{r'h_B}}(H_B)^2 \tag{31}$$

In Equations (29)–(31), the subscript $B$ indicates the value at the surf zone. As the phase function of $\phi$ is $x\left(\vec{x}, t\right) = s\left(\vec{x}\right) - \omega t$, the wave number obtained from the deformed mild slope equation can be expressed as follows:

$$\vec{k} = \nabla x = \nabla s \tag{32}$$

To obtain $|\nabla s|$ from Equations (25), (26) or (31), it is necessary to know the direction of the wave. There are only two equations to solve $a$, $|\nabla s|$, $\theta$ . The linearity of the wave phase function gradient is assumed to be irrotational by Equation (15), and the convergence conditions of the wave model are given by Equation (16).

$$\nabla \times (\nabla s) = 0$$
$$\nabla s = |\nabla s|\cos\theta\,\vec{i} + |\nabla s|\sin\theta\,\vec{j}$$
$$\frac{\partial}{\partial x}(|\nabla s|\sin\theta) - \frac{\partial}{\partial y}(|\nabla s|\cos\theta) = 0 \tag{33}$$

$$|H_{now} - H_{old}| \le \varepsilon_H(H_{now}) \quad , \quad \varepsilon_H = 0.001$$
$$|H_{1now} - H_{1old}| \le \varepsilon_k(H_{1now}) \quad , \quad \varepsilon_k = 0.001$$
$$|H_{2now} - H_{2old}| \le \varepsilon_k(H_{2now}) \quad , \quad \varepsilon_k = 0.001 \tag{34}$$

## 2.3. Particle Tracking Model

In this study, the particle tracking model is used to simulate the diffusion transmission. The model can be used to investigate the relationship between the influences of environmental factors on the suspended load and to understand the impact of changes in environmental factors caused by the construction of coastal structures on the transmission. The motion behavior of the simulated independent particle in the dynamic environment is an important tool for understanding the ocean pollution diffusion process, while the particle tracking model replaces the suspended load with the water quality point and treats

the diffusion process of the independent particle as random motion [19]. This study refers to the research of [20,21] from the power point of view:

$$Q_x = Q_c(u + U_r), \quad Q_y = Q_c(v + V_r) \tag{35}$$

$$Q_c = \frac{\left\{ A_1 f_c \left[ (u + U_r)^2 + (v + V_r)^2 \right] + A_2 \left( U_H^2 - U_{HC}^2 \right) \right\}}{g} \tag{36}$$

$$U_H = \sqrt{\frac{\tau_m}{\rho}} = \sqrt{\frac{f}{2}} U_{Max}, \quad U_{Max} = \frac{\pi H}{T \sinh(kh)} \tag{37}$$

where $u$ and $v$ are the velocities of the current in the $x$ and $y$ directions caused by wave motion; $U_r$ and $V_r$ represent the speed of the source in the $x$ and $y$ directions, respectively; $U_H$ is the maximum shear velocity caused by water particles on the seabed under the action of waves; $U_{Max}$ is the maximum velocity of water particles on the seabed under the action of waves; $T$ is the wave period; $H$ is the wave height; $h$ is the water depth; $f$ and $f_c$ are the friction coefficients of wave motion and average flow, respectively.

$$f = \begin{cases} 0.00251 \times \exp\left( 5.21 \times \exp\left( \frac{A}{k_s} \right)^{-0.19} \right) & , \quad \frac{A}{k_s} > 1.57 \\ 0.3 & , \quad \frac{A}{k_s} \le 1.57 \end{cases} \tag{38}$$

$$f_c = \frac{g}{C_c^2} \tag{39}$$

where $A_b$ is the half width of the wave orbit; $K_s$ is the bottom roughness; $U_{HC}$ is the critical shear velocity of water particles on the bottom. $U_{HC}$ refers to [22]

$$U_{HC} = 8.41 \times d_{50}{}^{11/32} \tag{40}$$

where $d_{50}$ is the average particle size on the bottom, which is between 0.565 mm > $d_{50}$ > 0.065 mm; $A_1$ is the sediment transport coefficient caused by the current; $A_2$ is the sediment transport coefficient caused by the wave motion.

In this particle tracking model, the vertical integral equations of the continuous equation and the equation of motion in the depth direction can be expressed as follows:

$$\frac{\partial \zeta}{\partial t} + \frac{\partial (HU)}{\partial x} + \frac{\partial (HV)}{\partial y} = 0 \tag{41}$$

$$\frac{\partial U}{\partial t} + U\frac{\partial U}{\partial x} + V\frac{\partial U}{\partial y} - fV + g\frac{\partial \zeta}{\partial x} + g\frac{U\left(U^2 + V^2\right)^{0.5}}{C^2 H} - \frac{1}{\rho H}\tau_x^s - \left( E_{xx}\frac{\partial^2 U}{\partial x^2} + E_{xy}\frac{\partial^2 U}{\partial y^2} \right) = 0 \tag{42}$$

$$\frac{\partial V}{\partial t} + U\frac{\partial V}{\partial x} + V\frac{\partial V}{\partial y} + fU + g\frac{\partial \zeta}{\partial y} + g\frac{V\left(U^2 + V^2\right)^{0.5}}{C^2 H} - \frac{1}{\rho H}\tau_y^s - \left( E_{yy}\frac{\partial^2 V}{\partial y^2} + E_{yx}\frac{\partial^2 V}{\partial x^2} \right) = 0 \tag{43}$$

In the two-dimensional mode, the position of each time step can be expressed as

$$X_n = X_{n-1} + U\Delta t \ ; \quad Y_n = Y_{n-1} + V\Delta t \tag{44}$$

where $n$ and $n-1$ each represent the old and new time steps; $u$ and $v$ are horizontal and vertical flow; and $\Delta t$ is unit time.

## 3. Model Set-Up

### 3.1. Modeling Procedure and Validation

Figure 2 displays a satellite image of the studied area, which was located at the offshore Changhua Coastal Industrial Park and had a water depth of −17 to −50 m. Figure 3 displays the calculation range of the model and the applied topographical data, which entailed a rectangular region with a length of 33 km and a width of 30 km, and the calculation ranged

from the south of Changhua Coastal Industrial Park in the north to Zhuoshui River in the south. The topography data employed by the model comprised the data measured in 2014 and 200 m water depth data from the National Science Council. Geographic data were collected through the following procedures: after the distribution of the plane wave field was calculated, the mechanisms of radiation stress and tide-level variation were applied to derive data regarding mixed wave fields, including inshore currents. Table 1 lists the configuration settings of the hydrodynamic model. The model was verified as described below.

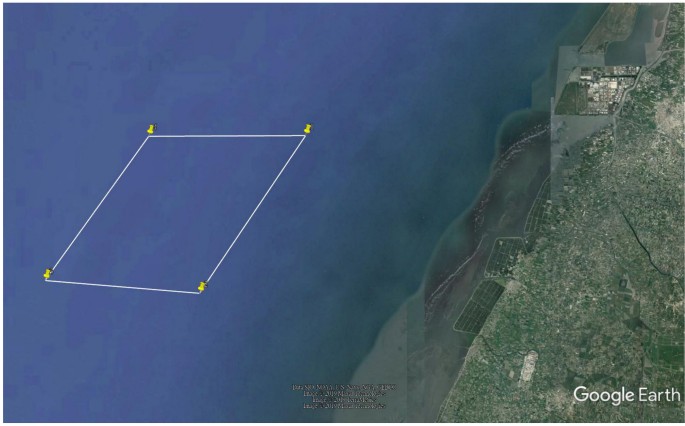

**Figure 2.** Satellite image showing the studied sea area.

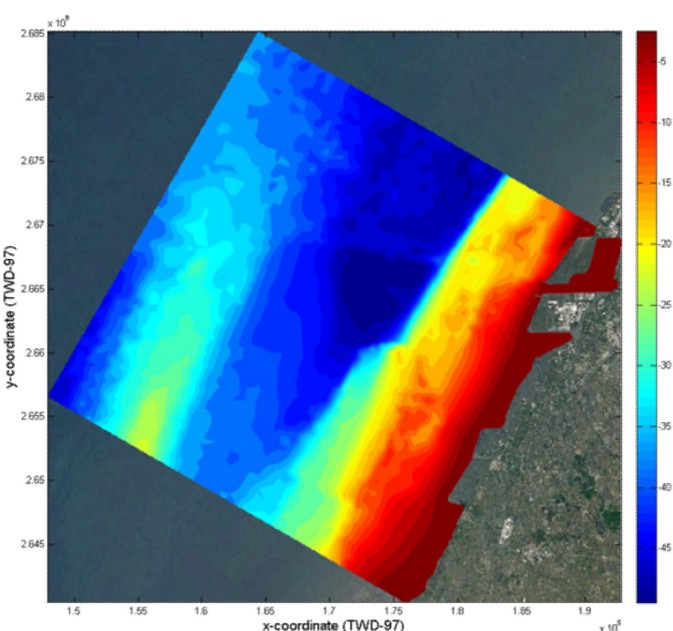

**Figure 3.** Topographical settings of the model.

**Table 1.** Calculation conditions for model calibration.

| Area | Grid Size | Number of Grid Points |
| --- | --- | --- |
| 33 × 30 km | 50 × 50 m | 660 × 600 |
| **Coordinate of the origin (TWD67)** | **Angle of deviation (counterclockwise from the north)** | **Time step size** |
| 147892, 2656583 | 25° | 1.5 s |

The currents simulated by the model generally flowed parallel to the coastline (Figures 4 and 5). Specifically, the current flowed from the southwest to the northeast during flood tides, and

from the northeast to the southwest during ebb tides. Figure 6 compares the model simulation results with the actual current measurement. The calculated inshore flow velocity, offshore flow velocity, flow direction, and changes in water level were all highly consistent with the measured data. The results, in accordance with the actual situation, indicate that the model successfully reproduced the characteristics of the current in the sea area adjacent to the study area. After the boundary conditions and related parameters adopted by the hydrodynamic model were determined, the model was employed to calculate the hydrodynamic and suspended load transport characteristics in different offshore wind turbines.

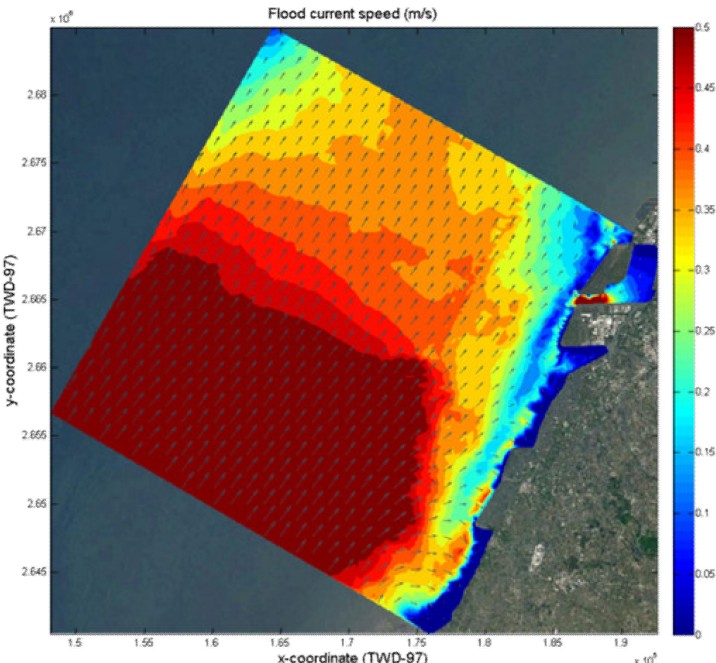

**Figure 4.** Current velocity and direction during flood tides.

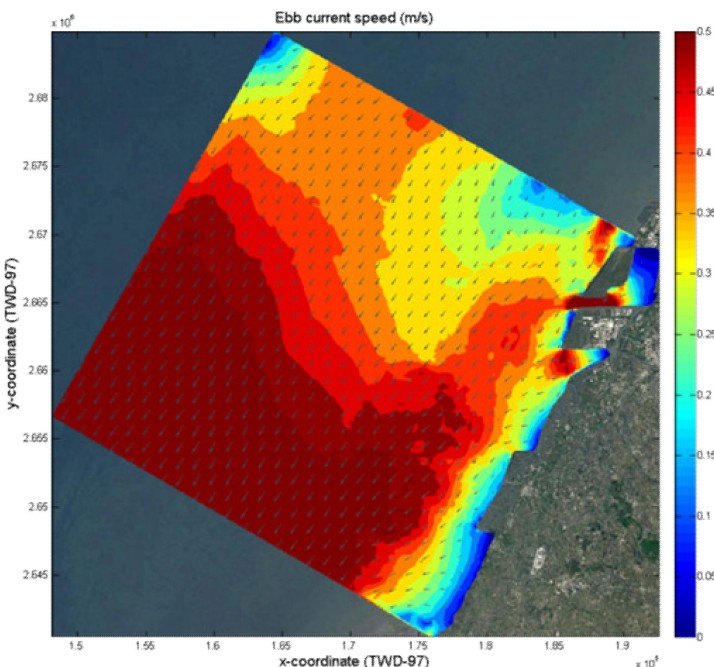

**Figure 5.** Current velocity and direction during ebb tides.

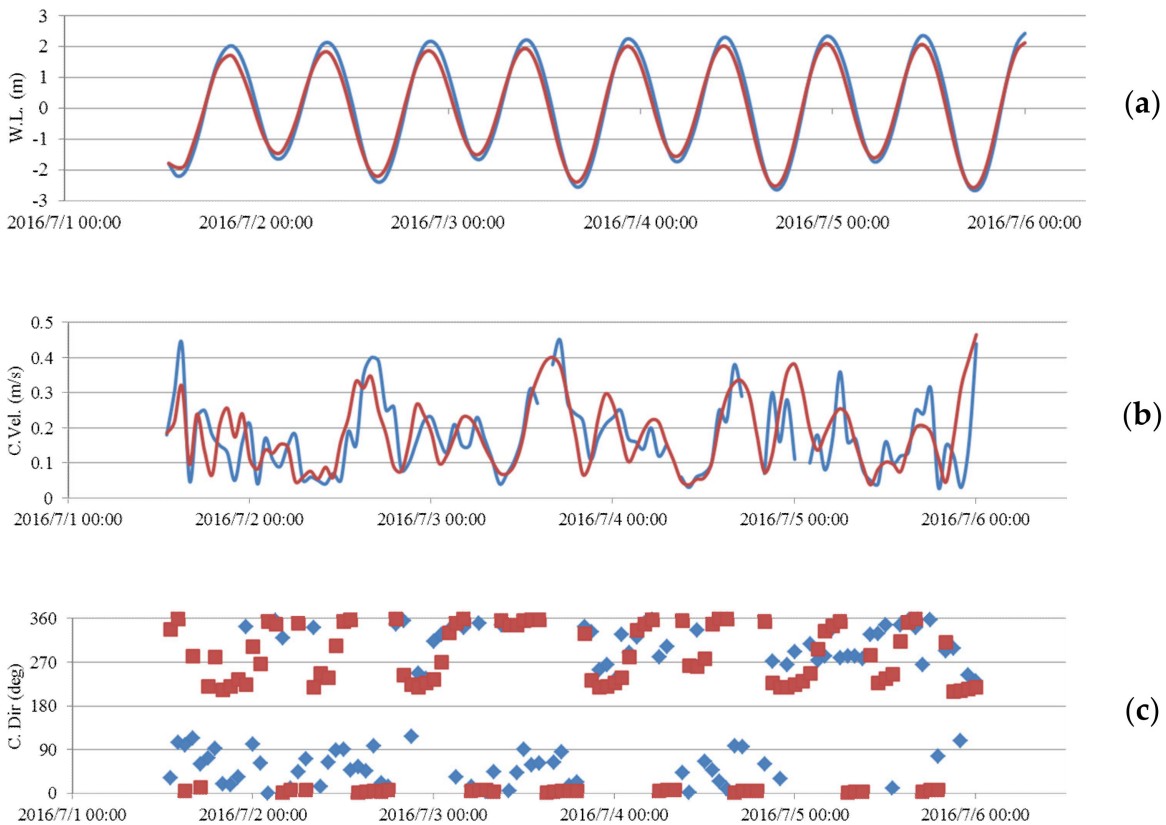

**Figure 6.** Comparison between the data obtained using the numerical model and the measured ocean currents: (**a**) tidal; (**b**) current velocity; (**c**) current direction.

### 3.2. Suspended Load Characteristics at the Different Foundations

The case study of the offshore wind turbines in this area is shown in Figure 7, and it used monopile, tripile, and jacket foundations. In the monopile, the pile diameter is 6 m; the diameter is 3 m in the tripile foundation, and the interval between piles is 30 m; the diameter is 3 m in the jacket, and the interval between piles is 20 m. The calculated water depth near the wind turbine foundation is approximately 20 m, and the simulated conditions were simulated in summer marine conditions (wave height below 0.5 m). We assumed that dredging and rubble bed trimming were performed for 8 h and that the per foundation setting operation was completed in 2 h. We referred to the conditions that have a large impact on pollution in the foundation construction, such as trailing suction hopper dredgers (1500 m³/h), and carried out continuous construction for analysis under the abovementioned working hours. The simulation results are shown in Figure 8. Take the wind turbine foundation as the center, and take the simulation results in four directions with a radius of 50 m. As this study is based on the assumption that dredging and rubble bed trimming were performed for 8 h and that the per foundation setting operation was completed in 2 h, the degree of influence on the tripile and jacket was larger than that on the monopile, and the influence time was longer. In Figure 8, the maximum value of the monopile is 0.012 kg/s, for the tripile, it is 0.018 kg/s, and for the jacket, it is 0.017 kg/s. The period of time to complete the construction of the maritime engineering was the time period with the highest concentration of suspended load. After that, because there was no source, the suspended load gradually spread, and the influence was gradually reduced. In addition, due to the influence of tidal currents on the sea, the suspended load also became more acceptable than the initial concentration.

**Figure 7.** Different foundation types: (**a**) monopile; (**b**) tripile; (**c**) jacket. The offshore wind foundation type images are from [23].

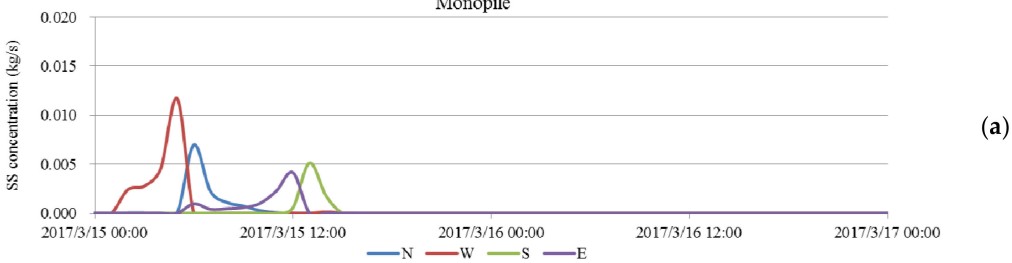

(**a**)

**Figure 8.** *Cont.*

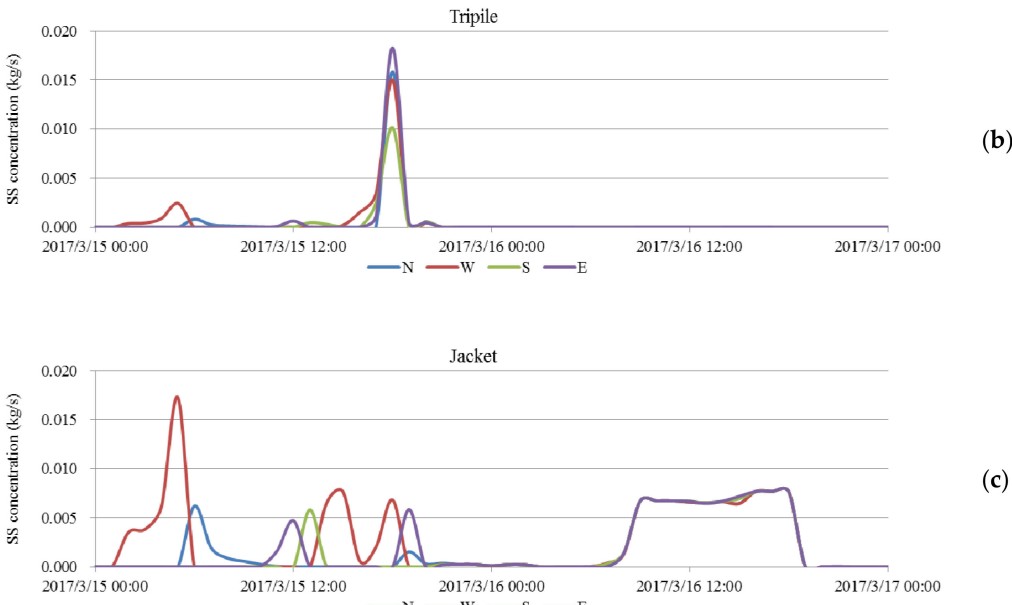

**Figure 8.** Simulation results of different foundation types: (**a**) monopile; (**b**) tripile; (**c**) jacket.

## 4. Conclusions

With different wind turbine foundations as a research subject, this study simulated the suspended load of summer conditions. The simulation indicated that the degree of influence on the tripile and jacket was larger than that on the monopile, and the influence time was longer. When the construction of the maritime engineering was completed, this period had the highest concentration of suspended load. After that, because there was no source, the suspended load gradually spread, and the influence was gradually reduced. In addition, due to the influence of tidal currents on the sea, the suspended load also became more acceptable than the initial concentration.

The validated model can be used to simulate the effect of suspended loads on offshore wind farms. In addition to attempting to understand the major suspended load mechanism and its impact within the studied area, subsequent studies can employ the current velocity and suspended sediment concentration during different return periods for scenario simulation. These should be implemented according to the analysis of trends in the targeted area under various circumstances.

**Author Contributions:** Conceptualization, H.-Y.W. and Y.-C.C.; methodology, Y.-C.C. and S.-S.H.; software, Y.-C.C. and H.-Y.W.; validation, S.-S.H., H.-Y.W. and Y.-C.C.; formal analysis, H.-Y.W.; investigation, H.-Y.W.; resources, Y.-C.C. and S.-S.H.; data curation, H.-Y.W. and Y.-C.C.; writing—original draft preparation, H.-Y.W. and Y.-C.C.; writing—review and editing, S.-S.H., H.-Y.W. and Y.-C.C.; visualization, Y.-C.C. and S.-S.H.; supervision, Y.-C.C. and S.-S.H.; project administration, Y.-C.C.; funding acquisition, Y.-C.C. All authors have read and agreed to the published version of the manuscript.

**Funding:** This research received no external funding.

**Institutional Review Board Statement:** Not applicable.

**Informed Consent Statement:** Not applicable.

**Data Availability Statement:** Not applicable.

**Conflicts of Interest:** The authors declare no conflict of interest.

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
