# Peer review of "A Study on the Influence of Suspended Matter by the Foundation Construction of Different Offshore Wind Turbines in the Taiwan Sea Area"

_water, doi:10.3390/w13101405_

Round 1

Reviewer 1 Report

The manuscript it is not yet ready for the publication. The following points deserve at least attention.

  1. State clearly the aim of the paper.
  2. Add the dphysical dimensions of the Chezy coefficient.
  3. How does the radiation stress enter the SW equations? If one follows what the authors write it is not possible to derive equations 2 and 3, because the radiation stress does not appear in the NS equation.
  4. How is the wave model linked to the calculation of the radiation stress? 
  5. How are the sediment particles considered? Are they massless particles? A concentration describes them? There is an equation for the concentration of suspended sediment? This is not said at all.
  6. Results are very poor. One cannot appreciate any difference among the different configurations shwon in fig. 6.  Fig. 7 is not comprehensible. Which is the information?
  7. Conclusion are very vague and poor. No quantitative information is given.

I cannot recommend the manuscript for the publication. The authors have to improve it before they can perform a new submission.

Author Response

We add description in the third paragraph of the introduction “In addition, study of the impact of offshore wind farm on marine ecology mostly focuses on the impact of fish, birds, and cetaceans, and there are relatively few studies on the impact of the marine environment and marine basic productivity. Among them, the three most important environmental factors affecting the primary productivity of the ocean are the abundance of light in the marine environment, the stability of water bodies, and the abundance of seawater nutrients. Due to the construction of offshore wind turbines may increase suspended load in the seawater and obscure the light in the sea, this effect may weaken the photosynthesis ability of phytoplankton in the seawater, leading to a decline in primary productivity. Accordingly, this study simulated the suspended load changes occurring after the installation of different wind turbine foundations. Numerical modeling was performed to examine the influence of the offshore wind farm set on its neighboring ocean environment. The results can serve as a reference for subsequent engineering construction activities and long-term geo-morphological change analysis.”

Then we check the wave, hydrodynamic and particle tracking models equations, and insert the image of the hydrodynamic model boundary condition. In addition, Figure 7 (the original Figure 6) is modified, and quantitative results are added to the description of the simulation results.

Reviewer 2 Report

The paper entitled 'A Study on the Influence of Suspended Matter by the Foundation Construction of Different Offshore Wind Turbines in the Taiwan Sea Area' deals with the important issue of studying the influence of foundation and type of wind turbine masts on the hydrodynamic conditions of the seabed. The paper is a simulation study comparing the impact of three structures: monopile, tripile, and jacket. Unfortunately, despite the interesting issue which the authors undertook, the paper is written in a very chaotic language. Already at the stage of introduction, it is difficult to find out what is the subject of the research. In the introductory part, there is a lack of unambiguously formulated objectives and discussion of results obtained by other authors. The authors use various computational models in their simulation studies. Unfortunately, due to a lack of literature references, it is not clear if some of them are own models or those of other researchers. There is also no information on how the calculations themselves were performed. It is not known whether the computational codes were used in-house or commercial. Comprehension of the article is made very difficult by numerous grammatical and stylistic errors. For this reason, the article, first of all, lends itself to a thorough linguistic improvement.

I do not belittle the amount of work and the topic that was undertaken. However, I would like to emphasize that the effort made by the authors deserves a much better description of the obtained results. By doing so, the work itself will gain in coherence and clarity of presentation. I recommend that, first of all, the paper should be very thoroughly improved from the linguistic point of view.

Detailed comments are given in the attached file.

Author Response

We have reviewed the manuscript we provided, and also used MDPI English editing service to revising grammatical and stylistic. In the introductory part, the study objectives are added description in the third paragraph. The computational models were used in-house computational codes, and we recheck the wave, hydrodynamic and particle tracking models.

Reviewer 3 Report

An interesting paper, dealing with the issue under investigation via a convincing methodology.

Overall a good effort, with just a few editorial improvements being highlighted in the associated pdf...

Author Response

Thank you, editor. We have reviewed the manuscript we provided. And also ask MDPI to conduct English editing service.

Round 2

Reviewer 1 Report

The manuscript has been improved, but the results section remained unchanged. It is quite surprising that after such a complicate development the most important result is given one page of comments. Moreover, how can the authors have results in kg/s if they copnsider massless suspended particles?

I am doubtful if this manuscript can be accepted or not. But leave the decision to the Editor.  I am not convinced. In case of acceptance, the authors should explain well their results and how from a lagrangian approach with massless particles they have derived the results shown in fig. 7

Author Response

Thank you for your comments, those comments are all valuable and very helpful for revising and improving our paper as well as the important guiding significance to our research. We have studied comments carefully and have made correction which we hope meet with approval. The main corrections in the paper and the responds to comments are as flowing:

The dynamic mechanism of suspend load is mainly caused by the action of waves and ocean currents. This study uses a hydrodynamic model that includes both waves and currents to establish a particle tracking model. The simulation results show the diffusion and transmission of suspend load, which is used to discuss the relationship between suspend load and the change of environmental factors, and to further understand the impact of environmental factors on suspend load caused by the changes in the foundation types of wind turbines.

Reviewer 2 Report

As suggested in the first review, the article entitled 'A Study on the Influence of Suspended Matter by the Foundation Construction of Different Offshore Wind Turbines in the Taiwan Sea Area' has been thoroughly revised linguistically. The most important parts of the paper have also been changed, including the description of the mathematical model used, thanks to which the text has gained clarity. In the revised version of the article, the authors took into account most, but not all, of the comments from the first review. As it stands, the submitted paper can be published in the Water Journal.

Author Response

We thank you very much for giving us an opportunity to revise our manuscript, we appreciate very much for their positive and constructive comments and suggestions on our manuscript. We have studied your comments carefully and have made revision. We have tried our best to revise our manuscript according to the comments. We would like to express our great appreciation to you for comments on our paper.